# Hyper-Skin: A Hyperspectral Dataset for Reconstructing Facial Skin-Spectra from RGB Images

**Pai Chet Ng** [1], **Zhixiang Chi** [1], **Yannick Verdie** [2], **Juwei Lu** [2], **Konstantinos N. Plataniotis** [1]

[1] The Edward S. Rogers Sr. Department of Electrical and Computer Engineering, University of Toronto

[2] Huawei Noah's Ark Laboratory, Huawei Canada

## Abstract

We introduce Hyper-Skin, a hyperspectral dataset covering wide range of wavelengths from visible (VIS) spectrum (400nm - 700nm) to near-infrared (NIR) spectrum (700nm - 1000nm), uniquely designed to facilitate research on facial skin-spectra reconstruction. By reconstructing skin spectra from RGB images, our dataset enables the study of hyperspectral skin analysis, such as melanin and hemoglobin concentrations, directly on the consumer device. Overcoming limitations of existing datasets, Hyper-Skin consists of diverse facial skin data collected with a pushbroom hyperspectral camera. With 330 hyperspectral cubes from 51 subjects, the dataset covers the facial skin from different angles and facial poses. Each hyperspectral cube has dimensions of $1024 \times 1024 \times 448$, resulting in millions of spectra vectors per image. The dataset, carefully curated in adherence to ethical guidelines, includes paired hyperspectral images and synthetic RGB images generated using real camera responses. We demonstrate the efficacy of our dataset by showcasing skin spectra reconstruction using state-of-the-art models on 31 bands of hyperspectral data resampled in the VIS and NIR spectrum. This Hyper-Skin dataset would be a valuable resource to NeurIPS community, encouraging the development of novel algorithms for skin spectral reconstruction while fostering interdisciplinary collaboration in hyperspectral skin analysis related to cosmetology and skin's well-being. Instructions to request the data and the related benchmarking codes are publicly available at: https://github.com/hyperspectral-skin/Hyper-Skin-2023.

## 1  Introduction

Hyperspectral imaging offers a comprehensive and non-invasive approach for facial skin analysis, capturing detailed spatio-spectral information across a wide range of wavelengths [1, 2]. This three-dimensional hyperspectral cube surpasses the limitations of single-point measurements, providing a deeper understanding of facial skin characteristics and spatial distribution [3]. Previous studies have demonstrated the potential of hyperspectral skin analysis in dermatology [4], cosmetics [5], and skin's well-being [6], paving the way for advanced analysis and applications in these domains. This paper introduces "hyper-skin", a hyperspectral skin dataset uniquely designed to facilitate the development of algorithms targeting on consumer-based cosmetology applications. This unique dataset is curated with this specific goal in mind, focusing its practical relevance within the consumer-based cosmetology and skin beauty.

Despite the potential of hyperspectral skin analysis on cosmetology and skin beauty, the high cost and limited accessibility of hyperspectral imaging systems have limited their widespread adoption. Consumer cameras, particularly those embedded in smartphones, have become an integral part of daily life and are extensively used for capturing selfies and everyday images. Hence, many works study the use of RGB images from consumer cameras for skin analysis [7, 8, 9]. While RGB

37th Conference on Neural Information Processing Systems (NeurIPS 2023) Track on Datasets and Benchmarks.

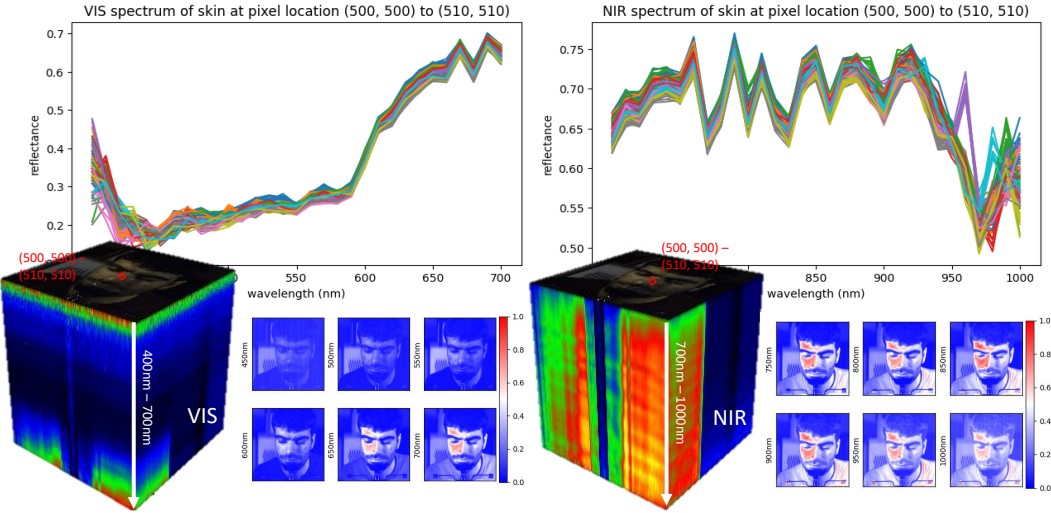

Figure 1: A glimpse of our Hyper-Skin dataset, covering the skin spectra in the visible spectrum (400nm - 700nm) and near-infrared spectrum (700nm - 1000nm).

images have been used for certain skin analysis tasks [10, 11], they lack the ability to capture the comprehensive spatio-spectral information provided by hyperspectral imaging, limiting the depth of skin analysis. In light of the prevalence of consumer cameras, an intriguing idea emerges: Can we reconstruct valuable information from expensive hyperspectral cubes using accessible RGB images, enabling hyperspectral skin analysis directly on consumer devices?

This highlights the need for a comprehensive dataset to develop computational reconstruction methods for the question above. While RGB datasets such as those from the International Skin Imaging Collaboration (ISIC) competition series (2016 - 2020) capture visual information, they lack the corresponding hyperspectral data required for studying hyperspectral reconstruction [12, 13, 14]. On the other hand, hyperspectral datasets enable the exploration of relationships between skin spectra and spatial distribution. Although the RGB counterpart can be synthetically generated from a given hyperspectral cube using a known camera response function, publicly available hyperspectral datasets focusing specifically on facial skin analysis are limited and often inaccessible to the public. Furthermore, existing hyperspectral datasets primarily focus on the visible (VIS) spectrum (400nm - 700nm), disregarding the valuable near-infrared (NIR) spectrum (700nm - 1000nm). These limitations highlight the necessity for a hyperspectral dataset that addresses these gaps and facilitates the development of low-cost and accessible hyperspectral skin analysis on consumer devices.

**Our Contributions**   Our Hyper-Skin dataset is uniquely designed to unlock the potential of hyperspectral skin analysis directly on the consumer device. With high spatial and spectral resolution, i.e., $1024 \times 1024 \times 448$, Hyper-Skin offers an extensive collection of hyperspectral cubes, yielding over a million spectra per image. Notably, we offer synthetic RGB images synthesized from 28 real camera response functions, allowing for versatile experimental setups. What sets Hyper-Skin apart is its comprehensive spectral coverage, including both the VIS and NIR spectrum, facilitating a holistic understanding of various aspects of human facial skin, enabling new possibilities for consumer applications to see beyond the visual appearance of their selfies and gain valuable insights into their skin's physiological characteristics, such as melanin and hemoglobin concentrations.

## 2   Related Work

The potential hyperspectral solutions in the skin-related analysis have encouraged the curation of hyperspectral datasets. This section reviews existing hyperspectral datasets related to skin analysis, as summarized in Table 2, and reconstruction aiming to provide affordable hyperspectral solutions accessible to consumers.

Table 1: Comparison between Skin-related Data

| Work | Experimental Subjects | Number of Subjects | Spectral Range | Spectral Resolution | Spatial Resolution | Acquisition Device |
|---|---|---|---|---|---|---|
| Ours | Full face | 51 | 400 - 1000 nm | 1.34 nm | 1024 x 1024 | Specim FX10 |
| [1] | Arm | 2 | 467 - 857 nm | 13 nm | NA | SkinSpect dermoscope |
| [2] | Facial skin | 204 | 400 - 700 nm | 3.3 nm | 1148 × 948 | SpectraCam® |
| [15] | Skin lesions | 330* | 465 - 630 nm | 10.3125 nm | 512 × 272 | Ximea MQ022HG-IM-SM4X4-VIS |
| [16] | Hairless mice | 10 | 500 - 660 nm | NA | NA | OxyVuTM -2, HyperMed T |
| [17] | Spine area | 17 | 450 – 950 nm | 12.2 nm | 500 × 250 | snapshot hyperspectral imaging (HSI) camera |
| [18] | Fore arm | 80 | 380 - 1055 nm | 2.8 nm | 450 × 1310 | Specim® Spectral Camera PS V10E |
| [4] | Hand | 45** | 397 - 1030 nm | 0.79 nm | 899 × 1312 | PFD-V10E line-scan camera |
| [19] | Skin reflectance | 144*** | 250 - 2500 nm | 5 nm | NA | NA |
| [9] | Mouse | 100** | 500 - 1000 nm | 5 nm | 640 x 480 | custom-built LCTF |

* Dermoscopy images, ** Reflectance images, *** Synthetic reflectance

**Skin-related Datasets**    SpectraCam and SpectraFace hyperspectral cameras [20] has been used to collect the data of normal and pathological skin with a spectral resolution of 31 wavelengths in the VIS spectrum [21]. The hyperspectral dermoscopy dataset consists of 330 images, including 80 melanoma images, 180 dysplastic nevus images, and 70 images of other skin lesions, with a spatial resolution of $512{\times}272$ pixels and 16 spectral bands ranging from 465nm to 630nm [15]. Hyperspectral dataset of 20 nude mice was collected by [16] as an alternative to human skin [22] to study acute changes in oxygenation and perfusion in irradiated skin. For markerless tracking in spinal navigation, [17] captured hyperspectral images of the skin from 17 healthy volunteers, with a spatial resolution of $1080{\times}2048$ and 41 spectral bands in the VISt to NIR range (450-950nm). Both work in [18] and [4] used the same Specim Spectral Camera PS V10E to acquire hyperspectral data covering the visible to NIR range (380-1055nm) with 1040 bands and a spatial resolution of $450{\times}1310$. The former dataset by [18], containing data from 80 subjects, focused on vein localization, whereas [4] used the dataset for a routine dermatological examinations. While references [18, 4, 17] involve capturing NIR spectral information, it's important to note that these datasets are not publicly accessible. Despite the potential of hyperspectral imaging in skin-related applications, most of these existing datasets relied on expensive imaging systems with a primary focus on scientific applications, and no efforts were made to provide low-cost hyperspectral solutions for consumer devices.

**Skin Spectral Reconstruction Datasets**    Although datasets specifically focused on skin spectra reconstruction are limited, there have been notable contributions in this field. One such dataset is the Skin Hyperspectral Reflectance Database (SHRD) [19], which includes 144 skin directional-hemispherical curves obtained through a novel hyperspectral light transport model. This dataset provides valuable insights into reconstructing hyperspectral information on human skin. Another study explores the reconstruction of hyperspectral information in mice skin [9], using 26 SKH-1 hairless albino mice as a model system [23]. The researchers propose a mathematical approach to reconstruct hyperspectral data from RGB images, allowing visualization of hemoglobin content across a large skin area without the need for expensive hyperspectral imaging systems. However, these skin-spectral reconstruction datasets have limitations in terms of sample size and spatial resolution, particularly in their coverage of facial images from various angles and poses. Consequently, progress in skin-spectral reconstruction has been relatively slower compared to hyperspectral reconstruction on natural scenes or everyday objects.

**Natural Scenes and Everyday Objects Reconstruction Datasets**    Compared to proprietary and unavailable skin-related datasets, several hyperspectral datasets on natural scenes and everyday objects have been made publicly available for the study of hyperspectral reconstruction. The CAVE dataset consists of 32 scenes with a spatial resolution of $512 \times 512$ pixels, 31 spectral bands ranging from 400nm to 700nm at 10nm intervals [24]. The HARVARD dataset includes 50 images captured under daylight illumination, with 31 spectral bands spanning from 420nm to 720nm, and a spatial resolution of 464x346 pixels [25]. The KAIST dataset comprises 30 hyperspectral images with a spatial resolution of $3376 \times 2704$ pixels and a spectral range from 420nm to 720nm [26]. The

KAIST-depth dataset features 16 indoor scenes with a spectral range of 420nm to 680nm, 27 spectral channels, and a spatial resolution of 2824 × 4240 pixels [27]. The New Trends in Image Restoration and Enhancement (NTIRE) series of datasets, including NTIRE2018, NTIRE2020, and NTIRE2022, have significantly advanced spectral reconstruction from RGB images, with varying sizes, spectral resolutions, and spectral ranges [28, 29, 30, 31]. Despite primarily focusing on natural scenes and generic objects, these publicly available datasets offer valuable resources, including facilitating the development of pre-trained models, that can be extended to skin spectral reconstruction tasks.

## 3 Hyper-Skin Data Curation and Preparation

This section outlines the methodology we employed for collecting facial skin data and provides a detailed description of our carefully curated Hyper-Skin dataset.

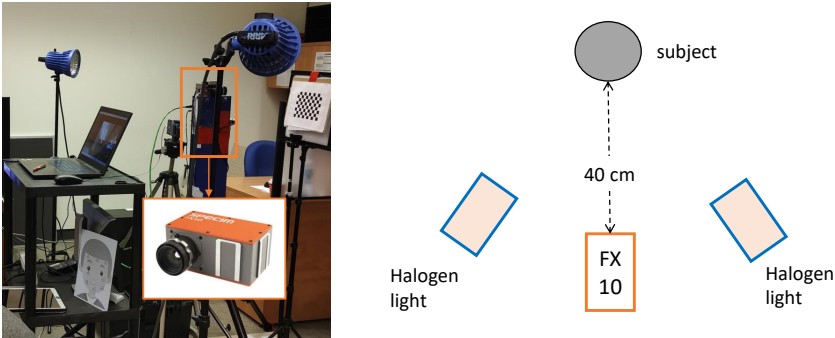

Figure 2: The image on the right illustrates our experimental setup for data collection, while the accompanying schematic representation provides a visual depiction of the setup.

### 3.1 Data Collection

The data collection process was conducted carefully, taking into account the setup of devices and recruitment of participants while ensuring adherence to the university's research ethics protocol. We successfully recruited 51 participants who contributed a total of 306 hyperspectral data. To maintain the privacy and sensitivity of the human subjects involved, we have implemented a credentialization procedure to access the dataset. Interested users will be required to digitally sign an End User License Agreement (EULA) online, which outlines the terms and conditions for using the dataset, including provisions for using only the authorized image in future publications. Detailed instructions for requesting the dataset will be publicly available in our GitHub repository, where users can find a digital EULA form to facilitate the data access request. Once the EULA form is signed and submitted, users will receive a secure link via email to download the data within 24 hours.

**Data Acquisition Devices and Set Up** The Hyper-Skin dataset was obtained using a Specim FX10 camera, covering 448 spectral bands from 400nm to 1000nm. Consider multiple factors, including participant safety, image quality, and spectral resolution, we opted to use a pushbroom camera rather than the Liquid Crystal Tunable Filter (LCTF) system used by [32]. The camera was moved using a customized scanner for precise scanning, as shown in Figure 1. The distance between the camera and the face was set at 40cm, providing a spatial resolution of 1024x1024 pixels. The scanner and camera were controlled by a computer running LUMO recorder software. Further setup details are available in the supplementary material. With a frame rate of 45Hz for one line, it took approximately 22.7 seconds to capture all 1024 lines. To minimize artifacts from line scanning, participants used a chin rest for stability. Halogen lamps illuminated the scene across the visible to near-infrared spectrum. Since ensuring participant safety (particularly for the eyes area) was a top priority, particularly for their eyes, the illumination level of the halogen lamps is carefully adjusted following the manufacturer's advice to prevent any risk to participants' eyes. Using the Specim FX10 camera resulted in high-quality images, setting our dataset apart from the mentioned dataset [32], which contains noisy and blurry images that affect skin texture visibility. Differing from the CMU dataset [32], which features a 10nm step size for 65 bands, our dataset encompasses both VIS and NIR spectrum with finer resolution.

Table 2: Hyper-Skin data pairs

| Description | (RGB, VIS) | (MSI, NIR) |
|---|---|---|
| Input | RGB | MSI (RGB + Infrared at 960nm) |
| Output | VIS (400nm - 700nm) | NIR (700nm - 1000nm) |
| Skin physiological features | surface-level characteristics (e.g., pigmentation and melanin map) | deeper tissue properties (e.g., collagen content and hemoglobin map) |

**Data Acquisition Process**  Participants were recruited through online forums and email advertisements, and their participation involved signing an informed consent form in accordance with the human research ethics protocol. The approved ethics protocol can be found in the supplementary materials. During the data acquisition process, participants were seated on a stool and asked to rest their face on a chin rest while maintaining stillness. Initially, participants were instructed to have a neutral facial expression, and three face images were captured from different viewpoints (front, left, and right) by rotating the chin rest. This process was then repeated with participants instructed to smile. A total of six images were collected for each participant. Throughout the camera scanning, a halogen light remained on. It's worth noting that even with minimal participant movement, slight shifting may occur as the FX10 camera scans line by line. To ensure high-quality images, the captured images were manually inspected by the investigator, and if any shifting was observed, the image was retaken until satisfactory results were achieved. Throughout the entire process, participant anonymity and confidentiality were strictly maintained.

**Participants Demographic and Cosmetology Condition**  Our data collection campaign attracted 51 participants, most participants are in their early 20s and 30s, with a smaller representation from other age groups (10s, 40s-50s). Male participants slightly outnumbered females, potentially due to the gender distribution in the Department of Electrical and Computer Engineering. The majority of participants identified as Asian, with a smaller number identifying as European or Latino. To improve the generalizability of our findings, we have applied for an extension of our research ethics protocol to conduct another data collection next year, aiming to include a more diverse sample.

## 3.2   Data Preparation

The Hyper-Skin dataset was created by collecting RAW hyperspectral data, which were then radiometrically calibrated and resampled into two separate 31-band datasets. One dataset covers the visible spectrum from 400nm to 700nm, while the other dataset covers the near-infrared spectrum from 700nm to 1000nm. Additionally, synthetic RGB and Multispectral (MSI) data were generated, including RGB images and an infrared image at 960nm. The Hyper-Skin dataset consists of two types of data: (RGB, VIS) and (MSI, NIR), offering different skin analysis capabilities. The visible spectrum data allows for the analysis of surface-level skin characteristics, such as melanin concentration, blood oxygenation, pigmentation, and vascularization. On the other hand, the near-infrared spectrum data enables the study of deeper tissue properties, including water content, collagen content, subcutaneous blood vessels, and tissue oxygenation. As summarized in Table 2, by providing these two distinct ranges of hyperspectral data, the Hyper-Skin dataset caters to different needs in skin analysis and facilitates comprehensive investigations of various skin features.

**Data Preprocessing**  We applied radiometric calibration on the RAW hyperspectral data to extract spectral reflectance information. This involved capturing a white reference image, representing a spectrally neutral surface with consistent reflectance values across all bands. A dark reference image was also obtained by closing the camera lens during capture. For precise calibration, selecting an appropriate white reference was crucial. After consultation with the camera vendor, we opted for cost-effective Teflon instead of Spectralon panels, as it provided satisfactory spectral response. The preprocessing steps included subtracting dark reference values to eliminate noise, and dividing by white reference values to normalize and convert data to reflectance values, yielding the desired spectral reflectance data.

**RGB and MSI data Generation**  The raw hyperspectral cube with 448 bands was resampled into two sets of 31-band data using SciPy's interpolation function. This downsampling to 31 bands

is in line with existing practices in hyperspectral reconstruction studies, similar to the CAVE and NTIRE2018-2022 datasets. It strikes a balance between data richness and size. This approach retains hyperspectral differentiation and computational efficiency for analysis, making the data more accessible compared to the 448-band dataset, which exceeds 1TB in size. While the complete 448-band dataset is substantial in terms of size, we are prepared to provide it upon specific request. The availability of the 31-band data addresses data transfer constraints, while the backup 448-band data ensures comprehensive access to the dataset.

For realistic RGB data generation, we adopted the HSI2RGB simulation pipeline based on ideal color-matching functions as outlined in [33]. Our emulation of consumer camera-captured images incorporates 28 camera response functions from [34] and [35], encompassing various cameras like DSLR and smartphones. Further details on the measurement setup and gathering camera spectral sensitivity information can be found in [34]. While we do possess actual RGB data captured by smartphone sensors, their current utility is constrained by concerns related to alignment quality issues stem from variations in camera models, viewing angles, and overlapping fields of view. In pursuit of maintaining high-quality aligned pairs, we have chosen to provide synthetic RGB images that are perfectly aligned with their hyperspectral counterparts. This approach aligns with existing practices such as those in the NTIRE2018-2022 challenges.

**Data Description** We intentionally chose 4 participants from the total of 51 to form the testing data. Each participant contributed 6 images, covering 2 facial expressions and 3 face poses. This selection ensured a comprehensive representation of facial poses and expressions in the testing dataset. The choice of these 4 participants was deliberate, based on their explicit consent for image use in publications, adhering to ethical standards. The remaining data was exclusively used for offline training. Collecting diverse participant images encompassed a variety of natural facial poses and expressions observed in selfies. Opting for a participant-specific approach, rather than random partitioning, prevents data overlap within the same participant in the training set, reducing potential bias. This strategic participant selection safeguards dataset integrity and subsequent analysis.

# 4 Evaluation and Benchmarks

This section discusses the benchmark design with the facial skin-spectra reconstruction and then presents the experimental results from both the spatial and spectral domains.

## 4.1 Facial Skin-spectra Reconstruction

The facial skin-spectra reconstruction task focuses on reconstructing the hyperspectral cube of the facial skin using the provided RGB image. Given a pair of RGB data represented as $R \in \mathbb{R}^{w \times h \times c}$ and the hyperspectral cube denoted as $H \in \mathbb{R}^{w \times h \times C}$, where $c << C$, the objective of the reconstruction task can be formulated as follows:

$$H = f(R; \Theta), \tag{1}$$

where the goal is to find the function $f(\cdot; \Theta)$ parameterized by $\Theta$ that maps the RGB data $R$ to the hyperspectral cube $H$. Given the extensive research on hyperspectral reconstruction for natural scenes or everyday objects, we can leverage existing hyperspectral reconstruction models as baseline models for our specific facial skin-spectra reconstruction problem.

### 4.1.1 Baseline Models

Numerous methods have been developed to address hyperspectral reconstruction from RGB images. Interested readers can refer to the survey by [36] for a list of representative models that have been developed over the past two decades. For our benchmark design, we specifically consider three models, i.e., Hyperspectral Convolutional Neural Network (HSCNN) [37], Hierarchical Regression Network (HRNet) [38], and Multi-stage spectral-wise transformer (MST++) [39], that have emerged as winners in the NTIRE competition series held in conjunction with CVPR from the year of 2018, 2020 to 2022, respectively.

#### 4.1.2 Evaluation Metrics

We consider two types of metrics: Structural Similarity Index (SSIM) for spatial evaluation and Spectral Angle Mapper (SAM) for spectral evaluation. Our emphasis lies in assessing the facial skin spectra reconstruction in human subjects, excluding the background image from analysis. This approach allows us to precisely gauge physiological properties like melanin and hemoglobin concentrations, essential for effective hyperspectral skin analysis.

Let $H \in \mathbb{R}^{w \times h \times C}$ represent the ground truth hyperspectral cube, and $\tilde{H} \in \mathbb{R}^{w \times h \times C}$ denote the reconstructed cube, where $C = 31$ for the 31-band data. In order to focus the evaluation specifically on the facial skin-spectra components within the hyperspectral cube $H$, we can utilize a mask $M \in [0, 1]^{w \times h}$ to exclude the background during the assessment. The mask $M(i, j) = 1$ indicates that the pixel at location $(i, j)$ corresponds to the human subject, while $M(i, j) = 0$ signifies the background pixels that should be discarded in the evaluation process. Let $H_s \in \mathbb{R}^{w \times h \times C}$ be the resulting matrix that has all the background removed. To obtain $H_s$, we need to compute the element-wise multiplication between the mask $M$ and $H$ at $C = k$, i.e.,

$$H_s(:, :, k) = H(:, :, k) \odot M \tag{2}$$

This element-wise multiplication is repeated for all channels of $H$ to obtain the final channel-wise multiplication matrix $H_s$.

**Spatial Evaluation**    The evaluation from the spatial domain focuses on the quality of the reconstructed cube at each band in terms of spatial similarity. For this, we use SSIM to compute the spatial similarity between the ground truth and reconstructed HSI. Let $h_s^{(P)}$ and $\tilde{h}_s^{(P)}$ be be the patch of the ground truth and reconstructed HSI, then SSIM between each patch can be described as follows:

$$SSIM(h_s^{(P)}, \tilde{h}_s^{(P)}) = l(h_s^{(P)}, \tilde{h}_s^{(P)})^\alpha c(h_s^{(P)}, \tilde{h}_s^{(P)})^\beta s(h_s^{(P)}, \tilde{h}_s^{(P)})^\gamma \tag{3}$$

where $\alpha, \beta$ and $\gamma$ are the weighting parameters for the luminance, contrast, and structural comparison functions. For the detailed formulation of each component, please refer to [40]. To account for mask-edge effects, if a patch is only partially covered by the mask, SSIM calculations consider only the masked pixels. This approach focuses evaluation on relevant information and minimizes the influence of background pixels on SSIM scores. However, in cases where a patch is entirely background, division by zero occurs in SSIM calculation. To avoid mathematical errors, such patches are excluded from evaluation. This ensures meaningful and reliable SSIM scores for informative patches within the mask.

**Spectral Evaluation**    On the other hand, the spectral domain evaluation aims to assess the accuracy of the spectral signature in the reconstructed cube at specific pixel positions. For this, we used SAM to compute the cosine angle between each spectra vector of the ground truth and reconstructed HSI. Let $\mathbf{h}_{ij} = H(i, j, :) \in \mathbb{R}^C$ be a $C$-dimensional spectra vector for the ground truth hyperspectral at location $i$ and $j$, and let $\tilde{\mathbf{h}}_{ij}$ be the corresponding spectra vector for the reconstructed cube, we can compute the SAM between these 2 spectra vectors as follows:

$$SAM(\mathbf{h}_{ij}, \tilde{\mathbf{h}}_{ij}) = \cos^{-1} \left( \frac{\sum_{k=1}^C \mathbf{h}_{ij}(k) \tilde{\mathbf{h}}_{ij}(k)}{\sqrt{\sum_{k=1}^C \mathbf{h}_{ij}(k)^2} \sqrt{\sum_{k=1}^C \tilde{\mathbf{h}}_{ij}(k)^2}} \right), \tag{4}$$

where $\mathbf{h}_{ij}(k)$ denote the pixel value at band $k$ for a spectra vector located at $i$ and $j$ of the HSI cube. Contrary to the perception that the cosine angle restricts similarity, it's important to clarify that SAM computes the inverse cosine, yielding a range of values between 0 and 3.142. This metric is chosen to quantify the spectral alignment between two spectra in an $n$-dimensional space, where the dimensionality corresponds to the number of spectral bands [41]. A lower SAM value signifies a better alignment with the reference (ground truth) spectrum.

### 4.2 Implementation Details and Experimental Results

We conducted the experiment using two datasets prepared in Section 3.2 and evaluated the performance of baseline models (MST++, HRNet, and HSCNN) [42], that were trained with the NTIRE2022 dataset licensed under the GNU General Public License v3.0 [43]. We then proceeded to re-train

Table 3: Spatial Evaluation with SSIM

| | | Pre-trained Models | | Re-trained Models | |
|---|---|---|---|---|---|
| | **Data** | (RGB, VIS) | (MSI, NIR) | (RGB, VIS) | (MSI, NIR) |
| **with Back-ground** | HSCNN [37] | $0.683 \pm 0.027$ | - | $0.916 \pm 0.013$ | $0.943 \pm 0.007$ |
| | HRNet [38] | $0.704 \pm 0.023$ | - | $0.933 \pm 0.021$ | $0.955 \pm 0.006$ |
| | MST++ [39] | $0.602 \pm 0.042$ | - | $0.923 \pm 0.011$ | $0.959 \pm 0.006$ |
| **w/o Back-ground** | HSCNN [37] | $0.816 \pm 0.021$ | - | $0.950 \pm 0.011$ | $0.964 \pm 0.006$ |
| | HRNet [38] | $0.813 \pm 0.023$ | - | $0.961 \pm 0.014$ | $0.971 \pm 0.005$ |
| | MST++ [39] | $0.766 \pm 0.035$ | - | $0.954 \pm 0.010$ | $0.974 \pm 0.004$ |

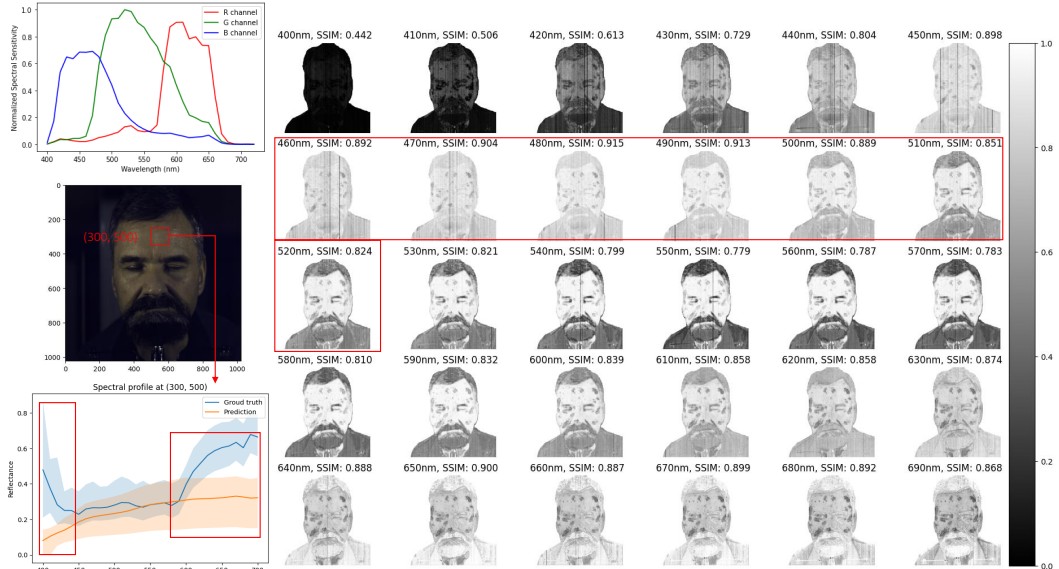

Figure 3: The top-right diagram represents the camera response function used to produce the RGB image. We conducted spectral evaluation across the 31 bands of images. The evaluation of the pre-trained HSCNN model's spectral performance reveals relatively better reconstruction in the middle bands (460nm - 520nm) as compared to the bands at the two ends. Notably, the spatial evaluation also shows that the skin areas in those middle bands exhibit better SSIM compared to the last few bands.

these models with our Hyper-Skin dataset for 100 epochs using an RTX5000 GPU and 16-bit memory. To address memory constraints, we randomly cropped the input size to 128x128 during training, while the entire 1024x1024 image was used for testing. The hyperparameters used for re-training the models remained the same, except for reducing the number of HSCNN blocks to 20 for the (MSI, NIR) experiments and adjusting the number of input channels to 4 for the (MSI, NIR) pair of data. Adam optimizer with a learning rate of 0.0004 was employed for training all three models.

### 4.2.1 Spatial Domain

Table 3 presents the spatial evaluation results using SSIM for two types of data: (RGB, VIS) and (MSI, NIR). The evaluation was conducted with and without the background, where the background was removed using a mask to focus on the human subject. Figure 3 provides a result illustration with pre-trained HSCNN. The pre-trained models were only applied to the (RGB, VIS) data pair since they were trained on the NTIRE dataset, which focuses on hyperspectral reconstruction in the visible spectrum. The (MSI, NIR) data pair requires an additional channel, which is not supported by the pre-trained models, thus they were not used for evaluation. After re-training the models with our Hyper-Skin dataset, significant improvements in performance were observed, as indicated in Table 3. Comparing the results with and without the background, it is evident that most of the reconstruction issues are associated with the background. For applications that focus on human skin, such as hyperspectral skin analysis, the performance of skin reconstruction is crucial. The

Table 4: Spectral Evaluation with SAM

| | Data | Pre-trained Models | | Re-trained Models | |
|---|---|---|---|---|---|
| | | (RGB, VIS) | (MSI, NIR) | (RGB, VIS) | (MSI, NIR) |
| **with Back- ground** | HSCNN [37] | $0.677 \pm 0.061$ | - | $0.119 \pm 0.008$ | $0.091 \pm 0.010$ |
| | HRNet [38] | $0.648 \pm 0.062$ | - | $0.147 \pm 0.014$ | $0.094 \pm 0.009$ |
| | MST++ [39] | $0.707 \pm 0.054$ | - | $0.113 \pm 0.009$ | $0.086 \pm 0.006$ |
| **w/o Back- ground** | HSCNN [37] | $0.621 \pm 0.049$ | - | $0.113 \pm 0.009$ | $0.083 \pm 0.012$ |
| | HRNet [38] | $0.596 \pm 0.046$ | - | $0.133 \pm 0.015$ | $0.086 \pm 0.010$ |
| | MST++ [39] | $0.628 \pm 0.050$ | - | $0.107 \pm 0.010$ | $0.076 \pm 0.005$ |

*Note: the pre-trained model based on NTIRE dataset is defined to take in the 3-channel input, and cannot be applied to MSI data that has 4-channel input.

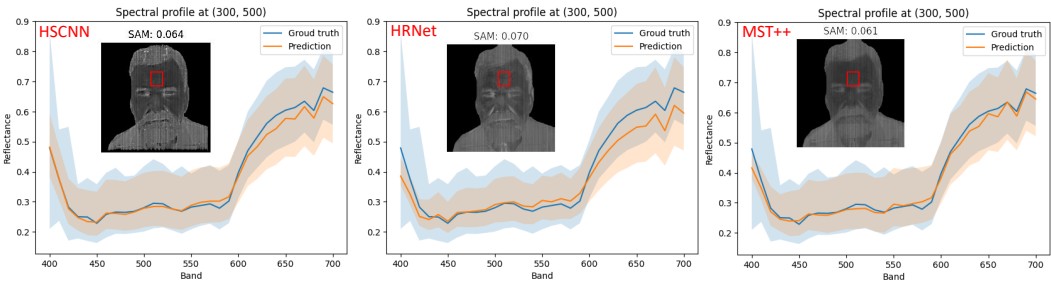

Figure 4: Re-training all three models with our Hyper-Skin dataset yielded a notable enhancement in performance, particularly in the skin area. However, it is worth noting that the first few bands of the reconstructed cube may not fully capture the variations present in the skin.

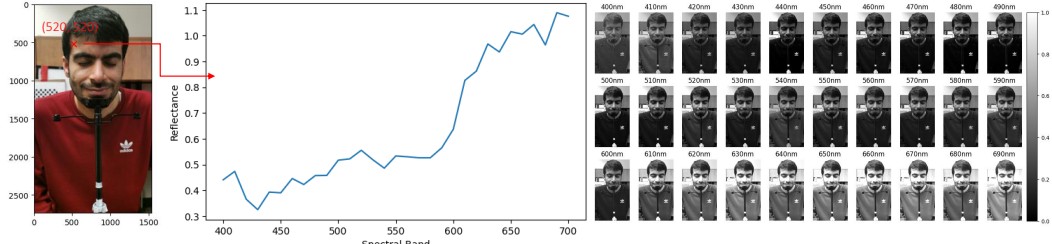

Figure 5: The demonstration of a real selfie image captured with a smartphone shows that the trained model is capable of reconstructing spectral information at the skin location.

results demonstrate that the reconstruction of the skin area exhibits better performance, supporting the potential for low-cost hyperspectral skin solutions, such as reconstructing the facial skin-spectra cube from smartphone selfies.

### 4.2.2 Spectral Domain

Table 4 shows SAM-based spectral evaluation results for two data types: (RGB, VIS) and (MSI, NIR), with and without background. SAM ranges from $\pi/2$ to 0, in which closer to 0 indicating better reconstruction. After re-training models using our Hyper-Skin dataset, both data types showed significant performance improvements, as illustrated in Figure 4. Note that the performance of the models was generally lower when the background was present compared to when it was removed. This highlights the impact of background interference on the reconstruction results. These findings emphasize the potential of our dataset in enhancing hyperspectral reconstruction for applications related to skin analysis and other fields that require accurate spatial information. Note that the re-trained models also showed good performance on the (MSI, NIR) data. We further verify the reconstruction performance on the real RGB image taken by a smartphone. As shown in Figure 5, the reconstruction model is capable of estimating the skin spectral information. Due to page constraints, the visualizations of these results are provided as supplementary materials.

# 5 Limitations, Ethical Considerations and Societal Impact

We worked closely with the university's ethical review board to ensure the inclusivity, privacy, and ethical integrity of the Hyper-Skin dataset. This ensured responsible data usage for skin analysis research. Our data collection followed strict participant consent procedures, including providing participants with detailed project information before obtaining written and verbal consent. The ethics review protocol and consent procedures are provided as supplementary materials.

**Limitations**   Our dataset's representativeness might be limited, potentially not capturing the full diversity of skin types, tones, and conditions across various populations. This could introduce biases and hinder model generalization. Additionally, while our deep neural network-based approach leverages the dataset for latent priors, it's constrained by dataset limitations. Novel cosmetology conditions not covered in training might affect model performance due to distribution shifts, a common challenge in machine learning [44, 45], particularly in medical datasets with skewed distributions [46, 47]. To address this, we've secured ethical approval to expand data collection, targeting diverse participants and cosmetology conditions to enhance practical utility.

**Ethical Considerations**   To ensure ethical compliance throughout the data collection process, we implemented measures to anonymize personally identifiable information, such as assigning a subject ID instead of using participants' real information. Robust security measures were also put in place to protect sensitive data from unauthorized access or misuse. We strictly followed guidelines provided by the research ethics board and obtained informed consent from every participant, respecting their autonomy and ensuring they understood how their data would be used.

**Societal Impact**   Our Hyper-Skin dataset revolutionizes skin analysis by providing affordable and accessible solutions. With its ability to reconstruct skin spectral properties and estimate parameters like melanin and hemoglobin concentration, it empowers researchers and practitioners to develop low-cost skin analysis solutions directly for consumers. The dataset's societal impact extends to individuals monitoring their skin's well-being and skincare companies developing personalized products and innovative AI models. By driving advancements in skin analysis, the Hyper-Skin dataset benefits individuals, professionals, and the skincare industry as a whole.

# 6 Conclusion

This paper contributes to the field of hyperspectral skin analysis by providing a comprehensive collection of facial skin hyperspectral data, named Hyper-Skin dataset. The novelty of this dataset lies in its spectral coverage in the VIS and NIR spectrum, offering potential applications in skin monitoring and customized cosmetic products at the consumer's fingertips. It serves as a valuable resource for algorithm development and evaluation, with future directions including dataset diversification, advanced analysis techniques, and interdisciplinary collaborations, inviting researchers and practitioners to contribute to the advancement of hyperspectral skin analysis for human well-being.

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
