# OpenReview forum: "Hyper-Skin: A Hyperspectral Dataset for Reconstructing Facial Skin-Spectra from RGB Images"
_NeurIPS.cc/2023/Track/Datasets_and_Benchmarks — NeurIPS 2023 Datasets and Benchmarks Poster_

### Official Review · Reviewer_CmSs · 2023-06-29
**Comprehensive facial hyperspectral image dataset**

**Rating:** 8
**Confidence:** 3
**Clarity:** The expression of this paper is clear.

**Strengths:**

The advanced aspects of this work are mainly reflected in the following areas:

1. It is a hyperspectral dataset specifically for facial skin spectral reconstruction, not a general dataset for natural scenes or everyday objects.

2. It is the first facial skin hyperspectral dataset that includes both visible light and near-infrared spectra, providing comprehensive information for studying surface and deep features of the skin.

3. It provides a facial skin hyperspectral dataset of synthetic RGB images and multispectral images generated by real camera response functions, promoting high-spectrum skin analysis on consumer-grade devices.

**Additional Feedback:**

This work is very useful for facial spectral reconstruction and high-spectral-based facial feature analysis. Currently, the author has implemented 3 advanced neural models on the dataset, and it is hoped that in the future more experimental results of traditional algorithms based on statistical or physical models can be released by the author, which will be very useful at a theoretical level.

**Correctness:**

I did not find any significant issues in the work.
The author used specialized equipment to ensure that the subjects remained still, and controlled the light source to cover all wavelengths. Finally, radiometric calibration and dark reference calibration were performed. The RGB images and multispectral images were generated using the response function of a real camera. I believe there are no problems with data collection.

**Documentation:**

The dataset has a hosting plan and access credentials are obtained by signing the EULA agreement, ensuring stable access to the dataset and preventing misuse.

**Ethics:**

The paper strictly adheres to ethical guidelines and protects the privacy of the subjects. Therefore, I have no ethical concerns.

**Limitations:**

The paper mentions the limitations of this work, which is good. The main limitation is that it cannot cover all types and conditions of skin. The author states that additional efforts will be taken to collect more samples from different ages, races, and regions.

**Opportunities For Improvement:**

This work is quite comprehensive, I only believe that there should be some additions in the experimental part. The author hopes to prove that this dataset can improve the performance of hyperspectral reconstruction, and compares models pre-trained on other datasets with those retrained on the Hyper-Skin dataset. SAM and SSIM show that by retraining on the Hyper-Skin dataset, it is possible to enhance model's performance for hyperspectral reconstruction. However, I think this comparison is not fair because it lacks cross-dataset comparisons. Training and testing only on the Hyper-Skin dataset cannot directly demonstrate its generalization ability. Therefore, I suggest adding cross-dataset experiments.

**Relation To Prior Work:**

This paper shows a significant improvement over previous work, as it includes hyperspectral data of skin in both visible light and near-infrared. This allows researchers to explore the optical characteristics of skin at different levels.

**Summary And Contributions:**

The main content of this article is to introduce a new hyperspectral dataset named Hyper-Skin, which is specifically used for reconstructing facial skin spectra from RGB images. Its main contribution is:

1. It covers a wide wavelength range from the visible (VIS) spectrum (400nm - 700nm) to the near-infrared (NIR) spectrum (700nm - 1000nm), promoting research on facial skin spectral reconstruction.

2. It provides synthetic RGB images and multispectral images generated by real camera response functions, along with paired hyperspectral images, enabling direct hyperspectral skin analysis on consumer-grade devices.

3. It contains 330 hyperspectral cubes from 51 subjects, captured from different angles and facial poses. Each cube has dimensions of 1024×1024×448, with each image containing millions of spectral vectors.

---

> ### Author Response · Authors · 2023-08-21
>
> Dear Reviewer CmSs,
>
> Thank you for your efforts in reviewing our paper and providing us constructive comments. We are pleased to receive your positive evaluation of our work and its contributions. Your suggested opportunity for improvement is duly noted. We have carefully considered your comments, and we would like to outline our responses to your feedback as follows:
>
> >**1. Cross-dataset experiments.**
> Thank you for your suggestions. We understand your concern about the lack of cross-dataset comparisons and the potential impact on demonstrating the generalization ability of our dataset. We agree that cross-dataset experiments could provide a more comprehensive evaluation. However, we would like to clarify the context in which our dataset was designed and the limitations in conducting such cross-dataset experiments.
>
> Our Hyper-Skin dataset is specifically tailored to hyperspectral imaging of human skin. It is unique in its focus on capturing the complex spectral characteristics of skin tissue. While cross-dataset comparisons are valuable for assessing generalization, the scarcity of publicly available hyperspectral datasets with a specific emphasis on human skin imaging limits our ability to conduct direct cross-dataset experiments.
>
> The primary goal of our work is to provide a specialized dataset that facilitates downstream skin analysis applications. Our dataset is designed to capture the intricate optical properties of skin in a controlled environment. As a result, the models trained on our dataset are optimized for hyperspectral analysis of human skin. Applying these models to datasets with different contexts, such as natural scenes or objects, may indeed result in distribution shift and reduced performance, as the models are specialized for skin.
>
> > **2. Statistical analysis / traditional methods / physical models.**
>
> Thank you for your positive feedback and valuable suggestion. We appreciate your recognition of the usefulness of our work in the domain of facial spectral reconstruction and high-spectral-based facial feature analysis. We are glad that you found our implementation of advanced neural models on the dataset valuable.
>
> We also acknowledge your interest in traditional algorithms based on statistical or physical models for facial spectral analysis. While our current focus has been on neural models due to their ability to capture complex relationships in the data, we agree that the inclusion of traditional algorithms would provide a more comprehensive analysis and contribute to the theoretical understanding of the field.
>
> We will consider your suggestion and explore the possibility of incorporating traditional algorithms in our future work.

---

> > ### Comment · Reviewer_CmSs · 2023-08-29
> >
> > Thank you for clarifying the limitations of this paper and future work.

---

### Official Review · Reviewer_5oGQ · 2023-07-21
**An interesting dataset on hyperspectral facial dataset for HSI feature reconstruction from RGB images**

**Rating:** 6
**Confidence:** 5
**Correctness:** Yes
**Clarity:** Yes

**Strengths:**

1. RGB data construction: Section 3.1 provides a clear and comprehensive explanation of the data collection methods and the procedure used for generating RGB/MSI data. The authors' detailed description allows readers to understand the data acquisition process thoroughly, ensuring transparency and reproducibility of their research.
2. Increased resolution and dataset size: One of the key strengths of their proposed dataset is its higher resolution and larger quantity compared to previous hyperspectral face datasets. This enhancement is a significant contribution to the field, as it allows for more detailed and comprehensive analysis of facial spectral information, offering researchers and practitioners a valuable resource for their studies.
3. Novel method for RGB/MSI data reconstruction: The approach used by the authors to reconstruct RGB/MSI data from HSI data is captivating and holds promise for practical applications. The decoupling of HSI data into (RGB, VIS) and (MSI, NIR) is an innovative concept that provides insights into the spectral characteristics of different facial features. This novel technique has the potential to advance research in the domain of hyperspectral imaging, opening up possibilities for various applications beyond facial analysis.

**Documentation:**

Yes

**Limitations:**

They clearly address the limitations of their work and provide future plans to cover the limitations.

**Opportunities For Improvement:**

1. Review of [r1] hyperspectral dataset: there already exists a dataset that presents hyperspectral images using a similar method to the current paper. Comparing and contrasting the two datasets would allow a better understanding of the novel contributions and unique features of the current paper.
2. Comparison of the dataset specification in a table: The reviewer suggests providing a clear and organized comparison of data specifications for existing datasets in a table format. Including information on sensor types, usage, and the number of images for each dataset, such as the existing dataset SHRD that covers a wider spectral range (400nm-2500nm), will aid readers in understanding the distinguishing characteristics of different datasets
3. Comparison with [21] dataset and contributions: As pointed out, [21] explores the roles of melanins, hemoglobins, lipids, and water in light attenuation within human skin across different spectral ranges. The authors should explicitly compare their dataset's spectral range and contributions to the findings in [21]. This comparison will help establish the significance and potential applications of the proposed dataset in skin well-being research, and how it complements or extends the insights from [21].
4. Participant size and bias removal: The reviewer raises concerns about the small number of participants (4) used in the experiments and the potential implications for bias removal and dataset generalization. The authors should address these concerns by discussing the limitations of the participant size and the measures taken to ensure reliable and unbiased results.
5. Impact of static illumination on results: The reviewer rightly points out that static illumination in the experimental environment might impact the results. The authors should acknowledge this limitation and discuss how they considered or controlled the effect of static illumination during data collection and analysis. This discussion would provide insights into the dataset's robustness and the validity of the findings.
6. Minor point: The reviewer observes a potential error in Eq. (4), where the presented method appears to be a cosine similarity, but the square exponent is only applied to the first term in the arccos function. The authors should clarify this equation to ensure its accuracy and coherence.

**Relation To Prior Work:**

See Weakness 1 and 3

**Summary And Contributions:**

The authors of this paper introduce an innovative and intriguing hyperspectral dataset called Hyper-skin, specifically designed for hyperspectral feature reconstruction from RGB data. This dataset comprises 330 hyperspectral cubes, each corresponding to 51 individuals. The spectral range covered by the hyperspectral sensor spans from 400nm to 1000nm. The concept of reconstructing facial spectral information from RGB data is captivating, and it holds significant potential for practical applications in real-world skin well-being and cosmetics. The ability to obtain spectral information from conventional RGB imagery opens up new possibilities for non-invasive and efficient analysis of skin conditions and cosmetic product efficacy. By providing this novel dataset, the authors contribute valuable resources to the hyperspectral imaging community, paving the way for further research and advancements in the field of hyperspectral feature reconstruction.

---

> ### Author Response · Authors · 2023-08-21
>
> Dear Reviewer 5oGQ,
>
> Thank you for your constructive comments. We are pleased to see your recognition of the innovative potential of our Hyper-Skin dataset and its implications for hyperspectral feature reconstruction. Your comments have greatly contributed to enhancing the quality and clarity of our work.
>
> In response to your comments, we would like to summarize the revisions we have made in our revised paper:
>
> >**1. There already exist similar dataset … compare and contrast …** \
> >**2. Comparison with other dataset (SHRD) using table … sensor types, usage, image numbers** \
> >**3. Compare with SHRD [21], spectral range and contribution**
>
> We appreciate the reviewer's suggestion to compare and contrast our hyperspectral dataset with the existing dataset presented in the paper [r1]. To address these related comments (2, 3), which pertain to dataset comparisons, we have streamlined them. In our revised manuscript, we have thoughtfully included a comparative analysis of various hyperspectral datasets, including ours, within a dedicated table labeled "Table 1: Comparison between Skin-related Data." This presentation aims to provide a clear and well-organized overview of the distinct characteristics of these datasets.
>
> Additionally, we appreciate your keen interest in understanding how our Hyper-Skin dataset compares to the dataset discussed in reference [21]. Reference [21] primarily focuses on dimensionality reduction of hyperspectral reflectance databases while maintaining fidelity during data reconstruction. This dataset is synthetically generated using the HyLIoS hyperspectral light transport model, to obtain the reflectance curve for various skin types and conditions, considering key parameters like eumelanin, pheomelanin, dermal blood content, water content, and lipids content. The synthetic dataset covers  a broader spectral range from 400nm to 2500nm for 144 distinct skin directional-hemispherical curves.
>
> While [21] presents a valuable contribution to hyperspectral data, there are differences in the objectives and methodologies between their work and ours. The primary goal of [21] is to investigate dimensionality reduction techniques and maintain fidelity during data reconstruction. In contrast, our Hyper-Skin dataset is uniquely designed for hyperspectral feature reconstruction from RGB data, particularly tailored for cosmetology applications and skin beauty assessment directly on consumer devices.
> Our Hyper-Skin dataset offers several advantages, including a direct imaging approach on human faces, higher spatial resolution, and a more extensive dataset size, with 330 hyperspectral cubes containing 448 spectral bands and a resolution of 1024x1024 pixels.
> Since SHRD is a synthetic dataset, it contains no information about the camera model, and number of human subjects. This absence of information makes a direct comparison challenging. However, we have carefully detailed the specifications of our Hyper-Skin dataset, including the number of hyperspectral cubes, spectral range, spatial resolution, and number of subjects, to provide a clear understanding of our contributions and dataset characteristics. We hope this clarification demonstrates how our work stands apart in terms of objectives and applications, as well as the extent to which our dataset complements or extends the insights from [21].
>
> [21] T. F. Chen and G. V. Baranoski, “Effective compression and reconstruction of human skin hyperspectral reflectance databases,” in 2015 37th Annual International Conference of the IEEE Engineering in Medicine and Biology Society (EMBC). IEEE, 2015, pp. 7027–7030.
>
> >**4. Small number of participants used in experiments … bias removal and dataset generalization.**
>
> Thank you for raising concerns regarding the participant size in our experiments. We would like to clarify that while we used 4 participants for testing, the remaining 47 participants were utilized for training. Each participant contributed 6 distinct images, resulting in a dataset that includes 31 spectral bands and a spatial resolution of 1024x1024 pixels. Moreover, as mentioned in our response to Area Chair B3pU, we will conduct cross-validation to ensure the robustness of our methods and address potential bias.
>
> We acknowledge the limitations associated with a relatively small participant size and the implications for dataset generalization. The recruitment process for participants is indeed a challenging aspect due to the sensitivity of human faces. Despite this challenge, we have made efforts to enhance the diversity of the dataset by working closely with research ethics review board for on-going participants recruitment campaign. We appreciate your feedback and are committed to addressing these concerns to ensure the reliability and unbiased nature of our dataset.

---

> > ### Author Response · Authors · 2023-08-21
> >
> > >**5. Impact of static illumination on results**
> >
> > Thank you for highlighting the potential impact of static illumination on our results. We want to address this concern by providing insights into how we considered and controlled the effect of static illumination during the data collection and analysis process.
> >
> > In our data collection setup, the acquisition of hyperspectral images using the FX10 camera required substantial light intensity. To achieve this, we utilized two Halogen lights provided by the FX10 camera supplier, as shown in Fig. 2. These lights were essential to provide the required illumination for the camera to capture hyperspectral images accurately. It's important to note that participants attended the data collection sessions on different days and times, which naturally introduced variations in the ambient lighting conditions. Additionally, the nature of our data collection process, where each image capture took several minutes and involved manual inspection, further contributed to potential illumination changes.
> >
> > Furthermore, our dataset includes RGB images captured separately (as mentioned in our response to the Area Chair) under different illumination conditions. This diversity in illumination, combined with the availability of hyperspectral and RGB images, allows us to assess the impact of static illumination on our results and address its potential effects. We appreciate your observation and are committed to discussing and controlling such factors to ensure the robustness and validity of our findings.
> >
> > >**6. Correction on Eq. 4.**
> >
> > Thank you for bringing this to our attention, we have corrected it in the revision.

---

> > ### Comment · Reviewer_5oGQ · 2023-08-22
> > **My mistakes on [r1]**
> >
> > I sincerely apologize for neglecting to include the reference [r1] in my review. Although late, I am attaching the missing reference [r1] below:
> >
> > [r1] Louis J. Denes, Peter Metes, & Yanxi Liu (2002). Hyperspectral Face Database [White paper]. Carnegie Mellon University.

---

> > > ### Author Response · Authors · 2023-08-23
> > >
> > > Dear Reviewer 5oGQ,
> > >
> > > Thank you for providing the missing reference [r1]. We acknowledge some similarities between the data collection set up of [r1] and ours, and we have consulted our camera vendor when planning the set up for our data collection campaign. We chose the pushbroom camera over the Liquid Crystal Tunable Filter (LCTF) system due to its finer spectral resolution and significantly lower lighting power requirement, which ensures the safety of participants' eyes during data collection.
> > >
> > > Moreover, we would like to highlight several distinctive aspects that differentiate our work from the mentioned dataset:
> > >
> > > 1.	Image Quality: Our dataset offers high-resolution (1024x1024) spectral images with superior quality. In comparison, the CMU dataset contains images with a resolution of 640x480. Notably, some images in their paper appear to be impacted by noise and blurriness, leading to a loss of visible skin texture. In contrast, our dataset delivers high-fidelity images.
> > > 2.	Spectral Sampling: While the CMU dataset employs a 10nm step size resulting in 65 spectral bands, our dataset covers the entire visible spectrum with a finer spectral resolution. This comprehensive spectral coverage is crucial for accurate skin analysis.
> > > 3.	Head Poses and Facial Expressions: Unlike the CMU dataset, which predominantly captures participants in a single head pose and various lighting conditions, our dataset features a diverse range of head poses and facial expressions. This diversity enhances the practicality of our dataset for real-world applications.
> > > 4.	Background Removal: To provide focused data for improved measurements, we captured background-only images for each participant. These images allowed us to generate masks that isolate human faces, providing algorithms with precise data.
> > > 5.	Real High-Quality RGB Images: In addition to hyperspectral data, our dataset includes high-quality RGB images captured simultaneously using both a smartphone and the hyperspectral camera. These images serve as valuable supplements for various reconstruction and skin analysis algorithms.
> > >
> > > We have included a citation to [r1] in the experimental setup section and revised Section 3.1 paragraph “Data Acquisition Devices and Set Up” to compare our set up with reference [r1], along with an explanation of our considerations for the final experimental setup for data collection.

---

> > > > ### Comment · Reviewer_5oGQ · 2023-08-24
> > > >
> > > > Dear Authors,
> > > >
> > > > I am deeply impressed with the response of the authors and leave a comment.
> > > > First of all, thank you very much for providing professional and careful answers.
> > > > Although I belatedly informed [r1], the literature review that exceeded expectations surprised the reviewers.
> > > >
> > > > In conclusion, all confusion has been resolved, and reviewers also believe that the quality of the paper has improved significantly.
> > > >
> > > > I am very happy to be able to improve my score.
> > > >
> > > > - Lastly, although this won't improve the score, I'd like to see a bit more improvement in the readability of Table 1. Only "of" exists in the 4th column.
> > > >
> > > > good luck.

---

> > > > > ### Author Response · Authors · 2023-08-24
> > > > > **Official Comment by Authors**
> > > > >
> > > > > Dear Reviewer 5oGQ,
> > > > >
> > > > > We are pleased to hear that your concerns were addressed. We also appreciate the increased rating and your additional comments. We have revised Table 1 in the updated version to make it clearer and more readable.
> > > > >
> > > > > Best regards,
> > > > >
> > > > > Authors of "Hyper-Skin"

---

> > > > > ### Comment · Area_Chair_B3pU · 2023-08-29
> > > > > **Is your Rating updated?**
> > > > >
> > > > > Dear reviewer 5oGQ,
> > > > >
> > > > > From your comments (after reviews), I take it that you intended to update (increase) the rating for this submission. Can you please make sure that has been done (in the interface above, I believe I still see the original rating score).
> > > > >
> > > > > Thank you!
> > > > >
> > > > > Kindest regards,
> > > > > Jochen Weber (Area Chair B3pU)

---

> > > > > > ### Comment · Reviewer_5oGQ · 2023-08-29
> > > > > >
> > > > > > Dear Area Chair B3pU,
> > > > > >
> > > > > > Thank you for your kind comment.
> > > > > > I have checked the score by observing the contents in the Revision button.
> > > > > > The reviewer is sure that the rating score is improved from 5 to 6.
> > > > > >
> > > > > > Thank you
> > > > > >
> > > > > > Reviewer 5oGQ
> > > > > >
> > > > > > Below is the history of the review rating score.
> > > > > >
> > > > > > > Official Review of Submission429 by Reviewer 5oGQ
> > > > > > 24 Aug 2023  (**revised**)
> > > > > > Title: An interesting dataset on hyperspectral facial dataset for HSI feature reconstruction from RGB images
> > > > > > Rating: 6: Marginally above acceptance threshold
> > > > > > Confidence: 5: The reviewer is absolutely certain that the evaluation is correct and very familiar with the relevant literature
> > > > > >
> > > > > > > Official Review of Submission429 by Reviewer 5oGQ
> > > > > > 21 Jul 2023  (**original**)
> > > > > > Title: An interesting dataset on hyperspectral facial dataset for HSI feature reconstruction from RGB images
> > > > > > Rating: 5: Marginally below acceptance threshold
> > > > > > Confidence: 4: The reviewer is confident but not absolutely certain that the evaluation is correct

---

### Official Review · Reviewer_rZAx · 2023-07-22
**Contributions towards skin spectra analysis**

**Rating:** 7
**Confidence:** 3
**Correctness:** See Limitations.
**Clarity:** Yes.

**Strengths:**

- Provision of publicly accessible high spatial and spectral resolution dataset for hyperspectral skin analysis
- Precise and clear dataset curation process

**Additional Feedback:**

- As mentioned by the authors in lines 70 to 74, studies like 3, 19, and 20 have captured NIR spectral information. This contradicts their statement in lines 46 and 47.
 - Based on the information provided in line 209, the Spectral Angle Mapper (SAM) is for the spectral domain. However, in line 262 and the caption of Table 3, it is mentioned as spatial evaluation using SAM, which creates ambiguity and confusion regarding its usage.

**Documentation:**

Yes.

**Ethics:**

To the best of reviewer's assessment, there are no obvious concerns. However, since the dataset takes into account the skin color, there may be concerns about adequate coverage of all skin types. That being said, authors clearly mention this as a limitation.

**Limitations:**

- As mentioned by the authors, the representativeness of the dataset is a potential limitation, as it may not fully encompass the diverse range of skin types, tones, and conditions present in different populations. This can introduce biases and hinder the generalizability of models trained on the dataset.
 - While the dataset includes 330 hyperspectral cubes, the number of subjects (51) is relatively small (with 4 participants chosen randomly for test set), which may limit the generalizability of the findings.

**Opportunities For Improvement:**

See Limitations

**Relation To Prior Work:**

Adequately covered.

**Summary And Contributions:**

The paper presents the Hyper-Skin dataset, a collection of 330 hyperspectral cubes obtained from 51 subjects, captured from varying angles and facial poses. The dataset's primary objective is to enable the reconstruction of skin spectra from RGB images taken with consumer cameras. It encompasses a broad range of wavelengths, spanning from the visible (VIS) to the near-infrared spectrum (NIR), allowing for the analysis of melanin and haemoglobin concentrations. The authors demonstrate the dataset's capabilities by reconstructing skin spectra using advanced models. They conducted experiment using (RGB, VIS) and (MSI, NIR) datasets and evaluated the performance of baseline models (MST++, HRNet, and HSCNN) obtained on the NTIRE2022 dataset by using Structural Similarity Index (SSIM) for the spatial domain and Spectral Angle Mapper (SAM) for the spectral domain. Their intention is to provide a resource (accessible hyperspectral dataset) that facilitates the development of algorithms for skin spectral reconstruction using the RGB image taken from consumers smartphones and encourages interdisciplinary collaboration in the field of hyperspectral skin analysis.

---

> ### Author Response · Authors · 2023-08-21
>
> Dear Reviewer rZAx,
>
> Thank you for your efforts in providing us the constructive comments and valuable feedback. We would like to take this opportunity to address the concerns you raised regarding the representativeness of our dataset, the potential limitations related to the number of subjects, and the 2 additional feedback:
>
> > **1. Representativeness of the dataset is potential limitation … 51 subjects and 330 hyperspectral cubes is limited.**
>
> Indeed, the representativeness of our dataset is a crucial consideration.
> We acknowledge that our dataset may not fully encompass the entire spectrum of skin types, tones, and conditions that exist across different populations, we are constantly working with the research ethics board to recruit more participants to enhance the diversity and inclusivity of our dataset.
>
> We would like to provide contextual information regarding the size and diversity of our dataset in comparison to existing hyperspectral datasets used in the NTIRE2018-2022 competitions held in conjunction with CVPR. Our dataset comprises 330 hyperspectral cubes, each with 448 spectral bands and a spatial resolution of 1024x1024. It's important to note that datasets such as the ARAD_1K dataset used in the NTIRE2022 challenge consist of 1000 images, with 31 spectral bands and a resolution of 482x512 of natural scenes and every day objects. Since our dataset is human-related, our data collection process was conducted with careful consideration. We engaged in recruiting participants, providing them with comprehensive information about potential risks, and obtaining their informed consent. During this process, it became evident that privacy concerns led some individuals to choose not to participate, which impacted the overall size of our dataset.
>
> > **2. the number of subjects (51) is relatively small (with 4 participants chosen randomly for test set), which may limit the generalizability of the findings.**
>
> In addition to the considerations of size and diversity, our data collection method also contributed to the relatively smaller number of subjects (51) included in the dataset. Our hyperspectral camera operates by scanning line by line, requiring careful manual inspection to ensure accurate alignment and quality. Images that did not meet our quality criteria were retaken, resulting in a time-consuming process for each capture. Despite these challenges, we made consistent efforts to enhance the dataset’s diversity by recruiting individuals from various demographics. Looking ahead, we are planning to collaborate with skin beauty industries to collect more data samples. This collaborative approach aims to address the limitation of a smaller subject pool and to improve the overall diversity of the dataset, contributing to the generalizability of our findings and applications.
>
> >**3. L70-74 contradicts the statement in L46-47**
>
> Thank you for pointing out this apparent inconsistency. We have revised the paragraph on “Skin-related Dataset” in Section 2 to clarify that the references [3, 19, 20] do indeed involve capturing NIR spectral information, but their datasets are not publicly available. Consequently, their limited accessibility means that their contribution to the broader research community might not be as extensive. Our intention was to emphasize the value of publicly accessible hyperspectral datasets, such as the one we provide with the Hyper-Skin dataset, to foster more inclusive and collaborative research efforts.
>
>
> >**4. Confusion on SAM:**
>
> Thank you for bringing this to our attention. We apologize for the confusion caused by the incorrect mention of SAM as a spatial evaluation method. The confusion arose from a typographical error, and we appreciate your understanding. We have corrected the errors in Section 4.2.2 and Table 3 to accurately reflect this fact.

---

### Official Review · Reviewer_cCX8 · 2023-07-22
**Although the collection of data was appropriate, the relationship between the data utilization plan and the analysis method presented in the paper was low.**

**Rating:** 6
**Confidence:** 3
**Clarity:** This document has been written to a m…

**Strengths:**

The strengths of this paper are (1) that it secured hyperspectral data from 400 nm to 1,000 nm, and (2) that it photographed the faces and expressions of real applicants from various angles.

**Additional Feedback:**

It would have been better if the data analysis for the various utilization methods suggested in the paper was included.

**Correctness:**

The data set was accurately measured including radiographic correction. However, I think the correlation with technology that analyzes human skin to derive various problems is low.

**Documentation:**

Yes, everything is included exactly.

**Ethics:**

This paper doesn't seem to be any problem.

**Limitations:**

The authors adequately addressed the limitations of the work and its potential negative social impact in the text and appendix.

**Opportunities For Improvement:**

Although the collection of data was appropriate, the relationship between the data utilization plan and the analysis method presented in the paper was low.

**Relation To Prior Work:**

Yes, Authors compared their work with previous work through references, and the differences were clearly discussed.

**Summary And Contributions:**

This paper built hyperspectral dataset covering wide range of wavelengths from visible (VIS) spectrum (400nm - 700nm) to near-infrared (NIR) spectrum (700nm - 1000nm). This Hyper-Skin dataset would be a valuable resource to NeurIPS community, encouraging the development of novel algorithms for skin spectral reconstruction while fostering interdisciplinary collaboration in hyperspectral skin analysis.

---

> ### Comment · Area_Chair_B3pU · 2023-08-09
> **Please add any suggestions for specific use cases you can think of**
>
> Dear authors of submission 429 (Hyper-Skin),
>
> Following up on this reviewers comments, I would encourage you to consider adding any specific applications for use cases (e.g., deriving proxy measure for metabolites/molecules/cell types present in the facial dermis) to the manuscript. I.e., if you can do a search of the literature in which other groups have (even if unsuccessfully!) attempted to use (hyper-) spectral imaging of (facial) skin to measure (or extrapolate) "skin health" (or other relevant metrics) and then mention those, this might increase the applicability of the manuscript for the NeurIPS audience at large.

---

> > ### Author Response · Authors · 2023-08-21
> >
> > Dear Area Chair B3pU,
> >
> > Thank you for the additional feedback. We have added cosmetology applications into our manuscript, focusing on use cases related to beauty assessment and personalized skincare. Our dataset's distinctive design is aimed at fostering the development of algorithms tailored to these specific applications, empowering users to harness the capabilities of hyperspectral imaging to enhance skin appearance and assist in making cosmetic-related choices.

---

> ### Author Response · Authors · 2023-08-21
>
> Dear Reviewer cCX8,
>
> Thank you sincerely for your comments. In our paper, we have outlined potential data utilization in Table 1 and provided an explanation of how the reconstructed hyperspectral data can be employed.  However, we acknowledge the necessity for a more comprehensive connection between data analysis techniques and their practical applications. To address this, we revised the Introduction by providing some examples that highlight the direct application of the reconstructed hyperspectral data in cosmetology-related contexts. For instance, by quantifying skin attributes such as melanin and hemoglobin concentrations, our dataset can facilitate the development of skin well-being and visual appearance assessment tools directly on consumer devices.
>
> We acknowledge that a comprehensive exploration of data analysis methods for various utilization scenarios would require an in-depth study that goes beyond the scope of this paper. However, we are actively working on further research to fully leverage the reconstructed data for skin analysis, taking advantage of the optical properties of skin-light interaction. This ongoing effort will enable us to extract essential physiological parameters from the hyperspectral data, contributing to the advancement of consumer-based cosmetology applications for skin beauty assessment.

---

### Author Response · Authors · 2023-08-21

Dear Area Chair and Reviewers,

We would like to express our sincere gratitude for your valuable time, effort, and insightful comments in reviewing our paper. Your constructive feedback has been instrumental in guiding our revisions. We have thoroughly revised the manuscript based on your comments and suggestions. In this revised version, we have addressed each of your comments with utmost care, ensuring that your insights are appropriately integrated into the content.

To facilitate your review, the tracked changes resulting from this revision have been appended to the main paper following the checklist. The changes have been highlighted for your convenience:
- Newly added content is designated with blue text.
- Deleted content is indicated with red strikethrough.

Lastly, we would like to extend our heartfelt appreciation for your commitment to the review process. We are eager to receive your feedback on the revised version and hope that our paper now aligns more closely with your expectations. Thank you.

\
Warm regards, \
authors of "Hyper-Skin"

---

### Decision · Program_Chairs · 2023-09-22

**Decision:**

Accept (Poster)

**Comment:**

Submission: Hyper-Skin (**#429**)
Bottom line: **accept (as poster)**

**Summary**: The authors collected novel data using a thoroughly described and in-house developed methodology: 306 images representing 3 viewing angle captures of human faces with two facial expressions were collected using a multi-spectral line-scanning camera. The final dataset is made available to the community as 1024x1024 pixel (original resolution) by 31-bands (downsampled) images, together with a number of assets that will allow a number of potential applications.

Pros: extending beyond existing work by (1) larger number of bands (and wider range of covered wavelengths), paired with (2) high-resolution standardized capture of human faces, as well as (3) benchmarking work on present dataset, (4) together with auxiliary assets (camera response functions, etc.); see particularly Table 1.

Cons: primarily (1) the small number of participants (selection bias), and (2) missing annotations, which makes it unlikely to support calibrating AI models for exploring specific applications (e.g., skin disease detection). In addition, (3) the exact applications are still somewhat vague (see however below).

Decision: As Area Chair, I am satisfied with the responses to the reviews, and I believe that the community will have to explore the dataset (as provided, and on request also the original 448-bands captures) to **discover applications** that the authors (and reviewers) probably cannot foresee. Given the primary nature of the data presented, I also believe that the data will find its use cases, and so support accepting this as a poster for NeurIPS 2023.

----

Reviewer cCX8: biggest downside mentioned was missing/unclear link between data and use cases; authors acknowledged and clarified in the manuscript -- at the same time, I (Area Chair) take it that applications will often require to "play with data" that is available, and so believe that with the revisions, the authors have provided sufficient motivation/incentive for the community to do exactly that.

----

Reviewer rZAx: biggest downside mentioned was/is lack of of skin type representation together with small sample size; authors acknowledge and then (unfortunately) **do not provide specific rebuttals**. It is my (AC, with now several years working as a data analyst in a dermatology department) belief that given the difficulty of this topic (racial diversity, representation of skin types), it would have been very difficult for the authors to provide a *scientifically and politically acceptable* argument that would not have required them to increase the dataset size by at least a factor of 2 or 3 (if not an order of magnitude). As such, I agree with the reviewer (broadly speaking), but yet believe that it is beneficial to publish the dataset **as is** for the benefit of the community; if the SAC/PCs would like to request that the authors add more language (beyond the second paragraph in the limitations section as it currently stands), I would support that, but do not believe this to be an impediment to publication -- with such a limitation clearly expressed! It is equally unfortunate that this reviewer did not provide a follow-up to the authors' (somewhat inconsequential) rebuttal, depriving me of any strong potential arguments against accepting the dataset **for this reason** at least...

----

Reviewer 5oGQ: biggest downside mentioned was/is the potential lack of novelty; I believe that while prior datasets each address one of the parameters the authors bring together (resolution, number of bands, image subject, etc.), the present dataset does, indeed, **bring these factors neatly together** in a single dataset (with the exception of being as large as a prior dataset with >200 subjects). All other concerns seem to be satisfactorily addressed.

----

Reviewer CmSs: highest rating (within the set of reviewers), and pretty much all (minor) concerns addressed.

----

AC concerns: the authors have addressed all concerns **to my satisfaction** (some concerns are limitations, which are equally addressed to my satisfaction).